# Malaria incidence rose following the introduction of neonicotinoid-based IRS in selected districts in northern Ghana: An observational analysis

Sylvester Coleman[1]*, Christian Atta-Obeng[2], Clinton Nkolokosa[3],
Abdul Gafaru Mohammed[2], Otubea Owusu Akrofi[2], Ihsan Isaka[2], Wahjib Mohammed[2],
Nana Yaw Peprah[2], Samuel Asiedu[4], Samuel K. Dadzie[5], Dominic B. Dery[6],
Julie-Anne A. Tangena[1], Adrienne Epstein[1,7], Keziah Malm[2]

1 Vector Biology Department, Liverpool School of Tropical Medicine, United Kingdom, 2 National Malaria Elimination Program, Ghana Health Service, Accra, Ghana, 3 Department of Vector Biology, Malawi-Liverpool Wellcome Programme, Blantyre, Malawi, 4 AngloGold Ashanti Malaria Control Limited, Accra, Ghana, 5 Noguchi Memorial Institute for Medical Research, University of Ghana, Legon, Accra, Ghana, 6 U.S. President's Malaria Initiative, U.S. Agency for International Development, Accra, Ghana, 7 Department of Medicine, University of California, San Francisco, San Francisco, California, United States of America

* sylvester.coleman@lstmed.ac.uk

## Abstract

In Ghana, indoor residual spraying (IRS) has been a key intervention for malaria control since 2008. After seven years of IRS with an organophosphate insecticide, which substantially reduced malaria incidence, IRS programs transitioned to neonicotinoid-based products between 2018 and 2019 as part of an insecticide rotation plan. This change was largely informed by entomological data from a limited number of pilot districts, with little evidence of epidemiological impact. We assessed the effect of this transition using monthly, district-level malaria incidence from 2015 to 2022 in 22 IRS districts across four northern regions. Incidence was calculated from routinely reported confirmed malaria cases in the Ghana Health Service District Health Information Management System, adjusted for testing rates, with population denominators from census-based estimates. An interrupted time series model with generalized estimating equations was fit to 2015–2019 data, when organophosphates were in use, to generate counterfactual trends through 2022, adjusting for rainfall, temperature, vegetation index, seasonality, and initiation of seasonal malaria chemoprevention. Observed incidence exceeded counterfactual estimates in the first transmission season after the switch and remained higher in subsequent campaigns. Across all districts, malaria incidence was 26% higher under neonicotinoid IRS than expected under continued organophosphate use (IRR = 1.26, 95% CI 1.07-2.08). Regional differences were observed, with significant increases in the Northern (IRR = 1.76, 95% CI 1.04-2.74), North East (IRR = 1.83, 95% CI 1.08-3.40), and Upper East (IRR = 4.26, 95% CI 2.17-8.50) regions, but not in the Upper West (IRR = 1.25, 95% CI 0.90-1.70). These findings suggest that the transition to neonicotinoids may

**Data availability statement:** All data supporting the conclusions of this article are included within the article and supplementary files.

**Funding:** SC is funded through the Wellcome Trust Accelerator Award [grant number 315444/Z/24/Z]. Funding for JAAT was provided by the Medical Research Council (MRC), United Kingdom [grant numbers MR/T031743/1 and MR/P027873/1]. SA is supported through the Global Fund Grant GHA-M-AGAMal: Building Effective Health Systems for Malaria Control – Ghana (01 January 2024–31 December 2026) awarded to AngloGold Ashanti Malaria Control Limited, which serves as an implementing partner for Indoor Residual Spraying (IRS) activities on behalf of the National Malaria Elimination Programme (NMEP), Ghana (https://data.the-globalfund.org/grant/GHA-M-AGAMal/4/over-view). AGAMal provided support in the form of salary for author SA. DBD was supported through the U.S. President's Malaria Initiative, U.S. Agency for International Development a main funder for IRS in Northern and North East regions of Ghana. USAID provided support in the form of salary for author DBD. The specific roles of these authors are articulated in the Author Contributions section. The funders had no role in study design, data collection and analysis, decision to publish, or preparation of the manuscript. The views expressed are those of the authors and not necessarily those of the Wellcome Trust, MRC, the Global Fund, AngloGold Ashanti Malaria Control Limited or the USAID.

**Competing interests:** Author SA is employed by AngloGold Ashanti Malaria Control Limited, an organisation implementing malaria control activities in Ghana supported by the Global Fund. DBD is also employed by the U.S. President's Malaria Initiative, U.S. Agency for International Development a main funder for IRS in Northern and North East regions of Ghana. These affiliations did not influence the study design, data collection and analysis, decision to publish, or preparation of the manuscript. This does not alter our adherence to PLOS ONE policies on sharing data and materials. The authors declare that no other competing interests exist.

have reduced IRS effectiveness in northern Ghana. Future pilots of new IRS products should incorporate both entomological and epidemiological outcomes to guide region-specific selection before national scale-up.

## Introduction

Indoor residual spraying (IRS) is an important tool for the control and elimination of malaria [1,2]. Increased investments in malaria control in the last two decades, including expanded IRS coverage, enabled the scale-up of vector control measures, leading to an estimated 37% decline in malaria incidence globally [3,4]. These gains have been reported across diverse geographical landscapes and against various mosquito vectors [5–9]. However, since 2018, the rate of decline in malaria burden has slowed in most countries, and some have even reported an increase in cases [10,11]. In Ghana, progress in reducing malaria cases has either stalled or reversed, with an estimated 5.3 million cases in 2022 compared to 5.2 million in 2018 [11].

A major challenge in reducing malaria is maintaining the efficacy of vector control tools, which is increasingly threatened by the complex and ever evolving resistance of mosquitoes to public health insecticides [12]. A direct consequence of resistance development in malaria vectors to existing control tools has been the need to shift towards insecticide products with alternative chemistries and modes of action [13]. Historically, vector control programs relied excessively on pyrethroids and carbamates for IRS [14]. However, IRS programs shifted to organophosphates in 2012 with the introduction of the microencapsulated formulation of Actellic 300CS (active ingredient [a.i.] pirimiphos-methyl) [15,16], and subsequently to neonicotinoids with the introduction of SumiShield 50WG (a.i. clothianidin) in 2017 and Fludora Fusion (a.i. clothianidin and deltamethrin) in 2018 [17,18]. In 2022, 11 out of the 13 sub-Saharan African countries supported by the President's Malaria Initiative (PMI) to implement IRS in their vector control programmes included neonicotinoids in their strategies [19]. Yet, there is still limited evidence supporting the effectiveness of these WHO-prequalified neonicotinoid-based formulations in suppressing malaria incidence, with initial field studies showing mixed results [20–22].

In Ghana, IRS has been a key component of the malaria control strategy since 2008 [23,24]. Whereas insecticide-treated nets (ITNs) have been distributed widely throughout Ghana, IRS has been centred in selected districts in northern Ghana where malaria burden is highest [25] and in Obuasi Municipal and Obuasi East District in the Ashanti region, where the AngloGold Ashanti Gold mine supports vector control activities. Since the switch in IRS insecticide formulations from pyrethroids to organophosphates in 2012, with a carbamate (Propoxur) used only in 2013, organophosphates (pirimiphos-methyl) remained the sole IRS insecticide until 2018, when neonicotinoids were introduced and gradually adopted [26]. A review of the introduction of non-pyrethroid- insecticides (primarily pirimiphos-methyl CS) in IRS programs across Africa revealed that the transition helped maintain program efficacy [27,28]. In Ghana, an observational analysis of routine health data demonstrated a significant

reduction in malaria incidence in IRS districts that switched from pyrethroids to organophosphates [7]. In 2018, neonicotinoid-based formulations were introduced in Ghana and had fully replaced organophosphates by 2021. This transition aligned with Ghana's insecticide resistance management plan, which promotes the rotation of insecticides with different modes of action to slow the development of resistance. As part of this strategy, the IRS program under the National Malaria Elimination Program (NMEP) piloted SumiShield 50WG in five districts. Based on residual efficacy results from this pilot and susceptibility data from multiple districts, neonicotinoid-based insecticides were adopted for broader IRS implementation starting in 2019. Although understanding the impact of this transition on malaria incidence is important, no epidemiological impact study has been conducted in Ghana to date. In this study, we assessed the impact of neonicotinoid-based insecticides on malaria incidence using monthly health facility data from 2015 to 2022 across 22 districts in northern Ghana.

## Materials and methods

### Study area

The study area comprises 22 districts in northern Ghana where IRS implementation shifted from organophosphate to neonicotinoid-based insecticides between 2018 and 2020 (Fig 1). These districts are located in the northern Savannah zone (comprised of the Guinea and Sudan Savannah), which is characterised by a relatively dry climate, with a unimodal rainy season that begins in May and ends in October, and precipitation ranges between 750 mm and 1,050 mm annually [29]. Across the study area, elevation ranges from approximately 100–500 m above sea level, with land cover mainly dominated by savanna woodland, interspersed with cultivated cropland, settlement, and water bodies [30,31].

Malaria is endemic and seasonal in the northern Savannah zone, with peak malaria transmission occurring between July and November [32]. The main vectors in the study area are *An. gambiae*, *An. coluzzii* and *An. arabiensis*. Resistance assays in the study area indicate high levels of pyrethroid resistance, while vectors have remained largely susceptible to both pirimiphos-methyl and clothianidin since their introduction [19,26]. An exception was observed in East Mamprusi District, where resistance to pirimiphos-methyl was detected in 2019, but susceptibility returned in subsequent years [26,33].

Vector control in our study area consists of IRS and ITNs. All districts received pyrethroid-only ITN during a mass ITN distribution campaign in 2016 [7]. From 2018 onwards, however, districts receiving IRS (the study districts) were exempted from subsequent mass ITN campaigns. ITN coverage in the study districts has, since 2018, been maintained primarily through continuous distribution channels, including antenatal clinics (for pregnant women) and child welfare clinics (for children under five years of age) [34]. Additionally, the intermittent preventive treatment of malaria in pregnancy IPTp-SP policy was adopted in 2003 as a national policy [35]. Seasonal malaria chemoprevention (SMC) was later introduced, beginning in the Upper West Region in 2015, followed by scale-up to the Upper East Region in 2016, and subsequently to the Northern and North East Regions in 2019. Since 2019, SMC has been implemented across all IRS districts as part of the broader malaria control strategy [36].

IRS in the Northern Region was introduced in 2008 in four districts, and until recently, with funding from PMI and the Global Fund, a total of 23 districts were sprayed annually. This included 11 districts in Upper West, five of six districts in North East, three of 15 in Upper East, and four of 17 in Northern Region. Thus, our study districts comprise 22 of the 23 IRS districts (Fig 1), with the exception of Tatale-Sangule, which has never received organophosphate-based IRS. These districts have historically been targeted for IRS because they were classified as areas of high malaria transmission.

A summary of insecticides used across the study districts is provided in Supplemental S1 Table, with formulation details in Table 1 and a timeline for the introduction of different interventions during the study period in Supplemental S2 Table. Most districts in the Northern and North East regions began IRS between 2008 and 2013, initially with pyrethroids, before switching to the organophosphate Actellic 300CS (AC) from 2012. In Upper East and Upper West, IRS started in 2012 with VectoGuard (a pirimiphos-methyl formulation), shifted to the carbamate ProGuard (propoxur) in 2013, and to Actellic 300CS in 2014. From 2018 onwards, there was a progressive shift to neonicotinoids: SumiShield 50WG (SS) and later

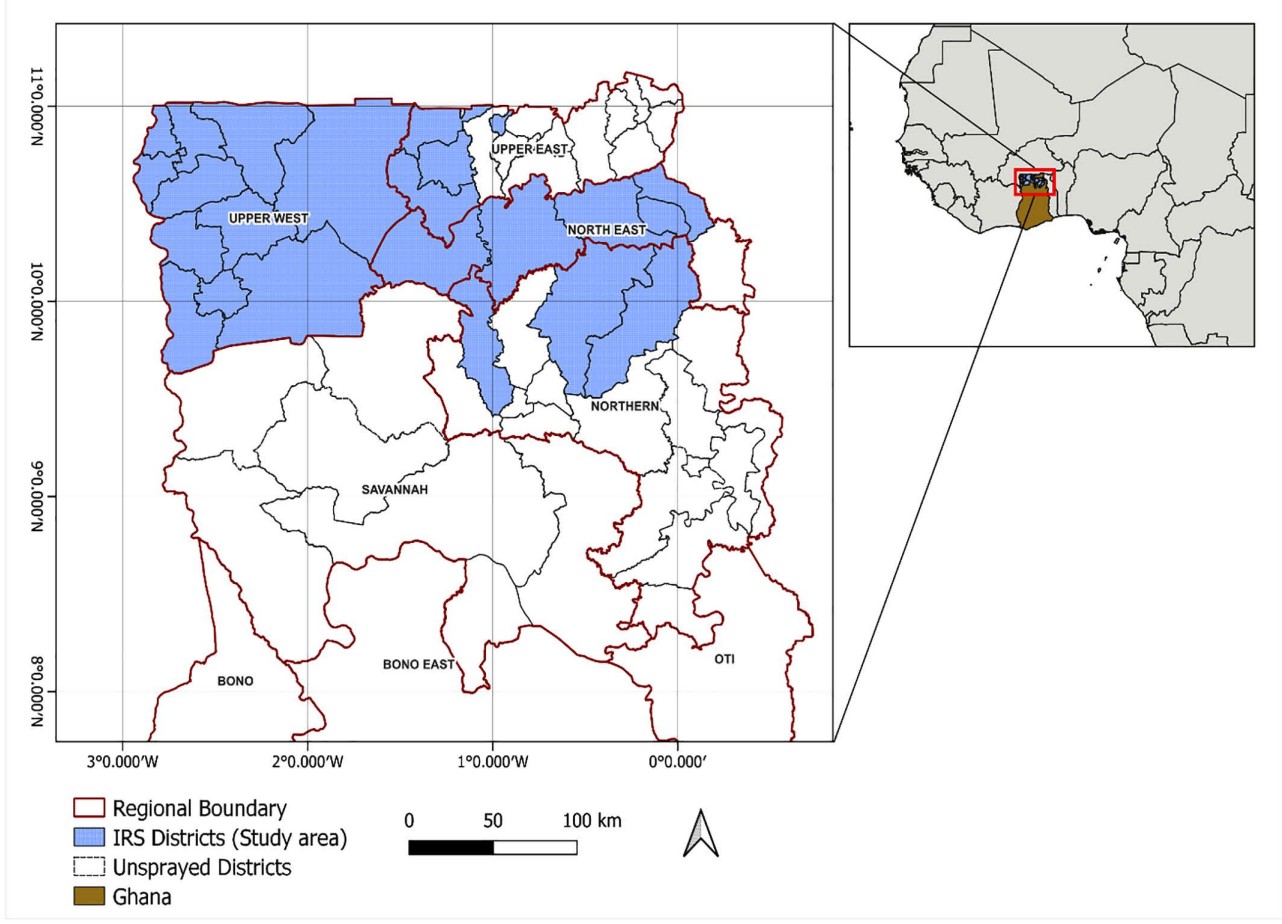

**Fig 1. Map of Ghana showing the study area (IRS districts -blue shading) across the Upper West, Upper East, North East, and Northern Regions of Ghana.** Red boundaries indicate regional administrative borders. The inset highlights Ghana's location within West Africa. Administrative boundaries were obtained from the Humanitarian Data Exchange (HDX), OCHA Field Information Services Section, Ghana administrative level 0–2 boundaries (COD-AB), sourced from Ghana Statistical Service (GSS) https://data.humdata.org/dataset/cod-ab-gha. Data are licensed under the Creative Commons Attribution 4.0 International (CC BY 4.0) licence. Map produced by the authors.

Fludora Fusion (FF). An exception was in the Mamprugu-Moagduri district, where spraying moved from Actellic to Sum-iShield, back to Actellic, and then to neonicotinoids (SS and FF). Monthly wall cone bioassays consistently showed that all IRS formulations maintained residual efficacy for over 10 months. Actellic 300CS (pirimiphos-methyl) remained effective for up to 11 months, while Fludora Fusion and SumiShield 50WG (clothianidin-based) also remained effective for at least 11 months across all surface types [37]. These residual profiles are sufficient to provide protection throughout the local malaria transmission season, typically lasting six-to-seven-months [38].

## Health-facility and environmental data

The primary outcome for this analysis was district-level malaria incidence, measured as cases per 1000 person-years, derived from routinely collected health facility malaria case data and adjusted for the testing rates. We estimated the malaria incidence using validated monthly passive malaria case data reported from 2015 to 2022 in the District Health Information Management System (DHIS2) of the Ghana Health Service. For each district and month, DHIS2 reports three

**Table 1. Type of IRS insecticide formulation used in our study districts from 2015-2022.**

| Region | Districts | IRS start year | 2015 | 2016 | 2017 | 2018 | 2019 | 2020 | 2021 | 2022 |
|---|---|---|---|---|---|---|---|---|---|---|
| North East | Bunkpurugu -Nakpanduri | 2011 | AC | AC | AC | AC | AC | SS | SS | FF |
| | East Mamprusi | 2009 | AC | AC | AC | AC | SS | FF | SS | SS |
| | Mamprugu-Moagduri | 2008 | AC | AC | AC | SS | SS | AC | FF | SS |
| | West Mamprusi | 2008 | AC | AC | AC | AC | SS | SS | FF | FF |
| | Yunyoo-Nasuan | 2011 | AC | AC | AC | AC | AC | FF | SS | SS |
| Northern | Gushiegu | 2008 | Nsp | Nsp | AC | AC | AC | FF | SS | SS |
| | Karaga | 2008 | Nsp | Nsp | AC | AC | AC | SS | SS | FF |
| | Kumbungu | 2008 | AC | AC | AC | AC | AC | FF | FF | SS |
| Upper East | Builsa North | 2013 | Nsp | Nsp | AC | AC | SS | SS | SS | FF |
| | Builsa South | 2013 | Nsp | Nsp | AC | AC | SS | SS | SS | FF |
| | Kasena-Nankana West | 2013 | Nsp | Nsp | AC | AC | SS | SS | SS | FF |
| Upper West | Daffiama-Bussie-Issa | 2012 | AC | AC | AC | SS | SS | FF | FF | SS |
| | Jirapa | 2012 | AC | AC | AC | SS | SS | FF | FF | SS |
| | Lambussie | 2012 | AC | AC | AC | AC | SS | SS | SS | FF |
| | Lawra | 2012 | AC | AC | AC | AC | SS | SS | SS | FF |
| | Nadowli-Kaleo | 2012 | AC | AC | AC | SS | SS | FF | SS | SS |
| | Nandom | 2012 | AC | AC | AC | AC | SS | SS | SS | FF |
| | Sissala East | 2013 | AC | AC | AC | AC | SS | SS | SS | FF |
| | Sissala West | 2012 | AC | AC | AC | SS | SS | FF | SS | FF |
| | Wa East | 2012 | AC | AC | AC | AC | SS | SS | SS | FF |
| | Wa Municipal | 2012 | AC | AC | AC | AC | SS | FF | FF | SS |
| | Wa West | 2012 | AC | AC | AC | AC | SS | SS | FF | FF |

Blue AC- the organophosphate Actellic 300CS (pirimiphos methyl), gold SS- the neonicotinoid SumiShield 50WG (Clothianidin only), green FF- the neonicotinoid Fludora Fusion WP SB (Clothianidin + Deltamethrin). NSp = not sprayed.

malaria indicators: the number of suspected malaria cases, the number of suspected malaria cases tested, and the number of confirmed malaria cases identified via microscopy or rapid diagnostic tests.

Malaria incidence was estimated as the number of confirmed malaria cases divided by the population at risk. Population values were estimated from the 2021 Population and Housing Census in DHIS2, adjusted for estimated annual growth. To account for temporal variation in diagnostic testing, confirmed case counts were adjusted for testing rates prior to analysis, using the proportion of suspected cases that were tested in each district and time period.

Data from the 2022 Demographic and Health Survey indicate that the proportion of febrile children for whom care was sought is notably high across the study area, with reported coverage of 75.3% in Upper West, 74.8% in Upper East, 71.7% in North East, and 62.2% in the Northern Region [39].

The period 2015–2022 was selected for this analysis because of the improved data quality assurance since 2015. DHIS2 data have historically been subject to quality concerns, including inconsistent reporting and delayed data entry. Analyses of malaria surveillance data from 2014–2017 have confirmed improved reliability of DHIS2 entries, particularly in IRS implementation areas [7]. Validation exercises conducted at the health facility level and supervised by District Health Management Teams have resulted in documented improvements in data completeness and internal consistency, with an accuracy of about 90% [40].

We adjusted for time-varying environmental variables that are associated with changes in malaria incidence, averaged at the district level. These include minimum and maximum monthly temperature in Celsius degrees [41], precipitation [42], Normalized Difference Vegetation Index (NDVI) [31], an indicator variable for calendar month (to adjust for seasonal effects),

and an indicator variable representing whether the district had initiated SMC. Temperature and precipitation variables were rescaled prior to analysis. Temperature was modelled per 10 °C increase and precipitation per 100 mm increase.

This work was conducted with permission and approval from Ghana's National Malaria Elimination Program as part of its mandate to review the impact of malaria interventions in Ghana.

## Statistical analysis

An interrupted time series (ITS) approach [43,44] was taken using R software and the *geepack* package [45] to assess the impact of the change in organophosphate to neonicotinoid-based IRS on malaria incidence. The following segmented regression model was specified:

$$Y_{ct} = \beta_0 + \beta_1 T + \beta_2 X_{ct} + \beta_3 TX_{ct} + \beta_4 R_{ct}$$

where $Y_{ct}$ is the outcome (malaria incidence) in district $c$ at month $t$, $T$ is the time elapsed since the start of the study in months, $X_{ct}$ is a dummy variable indicating the organophosphate-based IRS period (0) or neonicotinoid-based IRS period (1) for district $c$ at month $t$, and $R_{ct}$ is the vector of covariates for district $c$ at month $t$. As such, $\beta_0$ represents malaria incidence at the start of the study (t = 0), $\beta_1$ represents the change in malaria incidence associated with a one-month increase in the organophosphate-based IRS period, $\beta_2$ represents the level change in malaria incidence in the neonicotinoid-based IRS period, and $\beta_3$ represents the additional change in the slope after the change in IRS active ingredients. Generalized estimating equations were used to account for clustering at the district level. We used a Poisson distribution and modelled the count of malaria cases in district $c$ at month $t$, with an offset of the logged population denominator and an autoregressive order of 1 correlation structure to account for serial autocorrelation. The resulting segmented ITS model was used to estimate the counterfactual (unobserved) trend of malaria incidence in the absence of a change in IRS active ingredients (assuming that organophosphate-based IRS was continued) for each month by setting $X_{ct}$ to zero. Incidence rate ratios (IRR) were calculated by comparing the observed incidence to the counterfactual incidence, with bootstrapped 95% confidence intervals (CI). Models were first estimated across all districts. Subsequently, secondary analyses allowed the post-intervention slope change to vary by region by including a three-way interaction term between intervention, time since intervention, and region. This was motivated by the fact that in Ghana, health system structure and the delivery of IRS and other malaria control interventions are organised at the regional level, which could plausibly lead to regional differences in the temporal pattern of impact. We conducted an additional analysis to determine whether the impact of the change in the IRS active ingredient differed across transmission intensity (low transmission < 200 cases per 1,000 person-years during the baseline period; moderate transmission >= 200 cases per 1,000 person-years during the baseline period).

We conducted further sensitivity analyses to determine whether the results i) differed between the two clothianidin-based IRS products (SumiShield vs. Fludora Fusion) or depended on ii) how the data were selected, iii) the period considered, or iv) the way malaria cases were defined. First, we refit the interrupted time series model, replacing the single neonicotinoid-IRS-based indicator with a categorical variable for insecticide type (Actellic, SumiShield, Fludora Fusion) and included a time-since-spray variable that reset at the start of each IRS round. This allowed estimation of separate immediate-level changes and post-spray slopes for each product, accounting for sites that switched between products over time. In the second sensitivity analysis, we excluded districts that were not sprayed during the 2015–2016 IRS period. Third, we restricted the analysis to observations from 2017 onwards, when all the study districts were sprayed. Finally, we repeated the analysis using confirmed malaria cases as the primary outcome, without adjusting for testing rates.

## Results

Across the 22 study districts, an average of 46 months (range 27–62) of organophosphate IRS (OP-IRS) data and 44 months (range 34–57) of neonicotinoid IRS (NN-IRS) data were included in the analysis. Overall, the observed malaria

incidence averaged 233 per 1,000 person-years (range 75–369) during the OP-IRS period, compared to 298 per 1,000 person-years (range 95–480) during the NN-IRS period. Across most districts (18 of the 22 IRS districts), absolute malaria incidence was higher during the NN-IRS period compared to the OP-IRS period (Fig 2). The relative increases were most pronounced in the East Mamprusi, West Mamprusi, and Yunyoo-Nassuan districts in the North East Region (Supplemental S1 Fig). Major reductions in absolute malaria incidence were observed in only two districts - Wa Municipal (Upper West Region) and Kumbungu (Northern Region), while no change was recorded in Kasena-Nankana West and Mamprugu-Moagduri (Supplemental S3 Table).

Observed malaria incidence across all regions, stratified by region, showed marked seasonal peaks (Fig 3). Results from the ITS analysis show that in the months following the insecticide switch, observed incidence consistently exceeded the modelled counterfactual, which represents the expected incidence had organophosphate-based IRS been maintained. The divergence between observed and counterfactual trends was evident in the first transmission season post-switch and persisted across subsequent campaigns.

For all regions combined, the estimated IRR for the period after the switch was 1.26 (95% CI: 1.07-2.08). Regional differences were apparent (three-way joint interaction, p = 0.004). In the Northern, North East, and Upper East regions, incidence was significantly higher after the switch compared with the counterfactual (Northern: IRR = 1.76, 95% CI

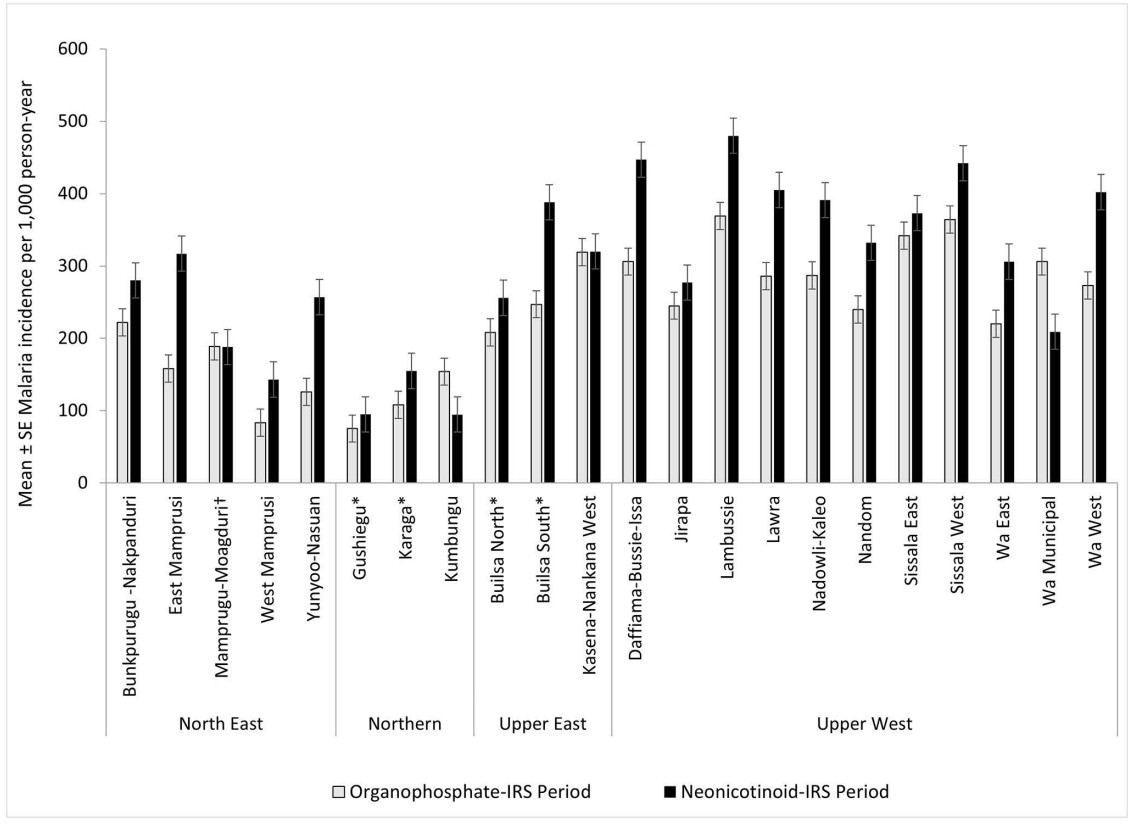

**Fig 2. Comparison of mean absolute malaria incidence (per 1,000 person-years) across districts, grouped by region, during periods when indoor residual spraying (IRS) was conducted with organophosphate (OP) insecticides versus neonicotinoids (NN).** Each pair of bars represents one district, with incidence rates from the OP-IRS period shown in grey and those from the NN-IRS period shown in black bars. *Observed start date for the period under consideration was January 2015, with an end date of December 2022 for all districts except Gushiegu, Karaga, Builsa North, Builsa South, and Kasena-Nankana West, for which the observed start date was January 2017. †In Mamprugu-Moagduri district, spraying shifted from Actellic to SumiShield, back to Actellic, and then to SumiShield and Fludora Fusion.

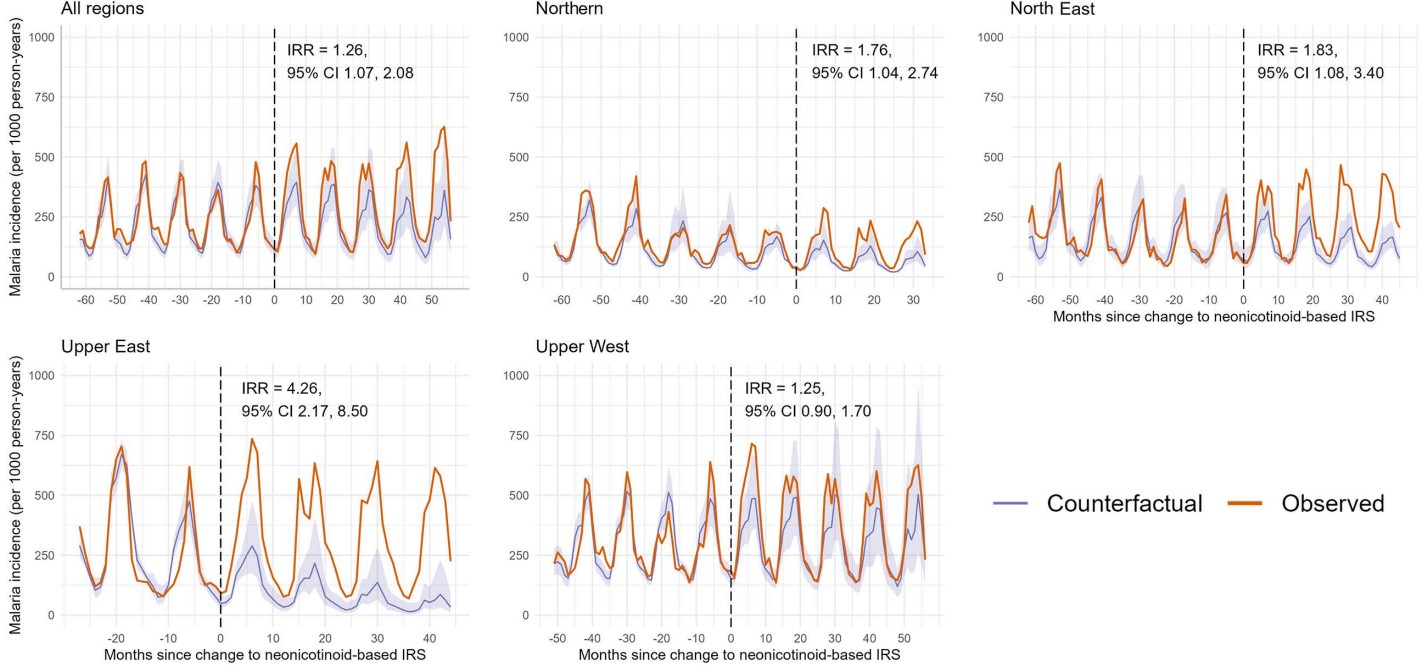

**Fig 3. Observed and modelled counterfactual monthly malaria incidence before and after the change to neonicotinoid-based IRS, overall and stratified by region.** Panels show observed (orange) and modelled counterfactual (purple with 95% CI) monthly malaria incidence per 1,000 person-years pooled across All Regions, as well as the individual regions (Northern, North East, Upper East, and Upper West). The dashed line at Month 0 marks the switch from organophosphate to neonicotinoid IRS. "Observed" refers to the actual reported malaria incidence from DHIMS2 data, while "counterfactual" represents expected incidence had organophosphate (pirimiphos methyl) IRS continued. IRRs (with 95% CIs) indicate the impact of the switch, adjusted for rainfall, temperature, NDVI, and initiation of seasonal malaria chemoprevention.

1.04-2.74; North East: IRR = 1.83, 95% CI 1.08-3.40; Upper East: IRR = 4.26, 95% CI 2.17-8.50). In contrast, in the Upper West region, no significant difference was detected (IRR = 1.25, 95% CI 0.90-1.70; Fig 3 & Supplemental S4 Table). When Mamprugu-Moagduri, where *Actellic 300CS* was reintroduced in 2020, was excluded from the analysis, malaria incidence increased substantially, from an overall regional incidence rate ratio (IRR) of 1.26 to 1.38 (95% CI: 1.12–1.92). Whilst the IRR in the North East region increased from 1.83 to 2.57 (95% CI: 1.97–4.00) when Mamprugu-Moagduri was excluded from the analysis (Supplemental S2 Fig). Covariate coefficients can be found in the supplemental S5 Table.

To further investigate regional differences, we assessed whether baseline transmission intensity modified the impact of switching to neonicotinoid-based IRS. The effect of the change differed by baseline incidence category (Fig 4; three-way joint interaction, p = 0.002). In districts with low baseline incidence, malaria incidence increased by 54% relative to the counterfactual after the switch (IRR = 1.54, 95% CI 1.14-2.23). In moderate-incidence districts, the increase was 20% (IRR = 1.20, 95% CI 1.02-1.43).

In sensitivity analyses that examined whether the results differed by the two neonicotinoid-based IRS products (Sum-iShield vs. Fludora Fusion), we found no evidence of a difference in immediate effect ($\chi^2(1)$=2.29, p = 0.13) or in post-spray slope ($\chi^2(1)$=0.03, p = 0.86) between the two formulations. Excluding districts not sprayed in 2015–2016 resulted in higher observed malaria incidence relative to the counterfactual following the intervention (incidence rate ratio (IRR) = 1.35, 95% CI 1.01–1.78; S3 Fig). When the ITS analysis was limited to data from 2017 onwards (when all the districts received continuous IRS), the same pattern was observed. Malaria incidence increased by about 21%, although with reduced precision (IRR = 1.21, 95% CI 0.90–1.85; S4 Fig). Using confirmed malaria cases only as the outcome yielded comparable estimates (IRR = 1.22, 95% CI 0.87–1.86; S5 Fig). Across all sensitivity analyses, observed malaria incidence consistently

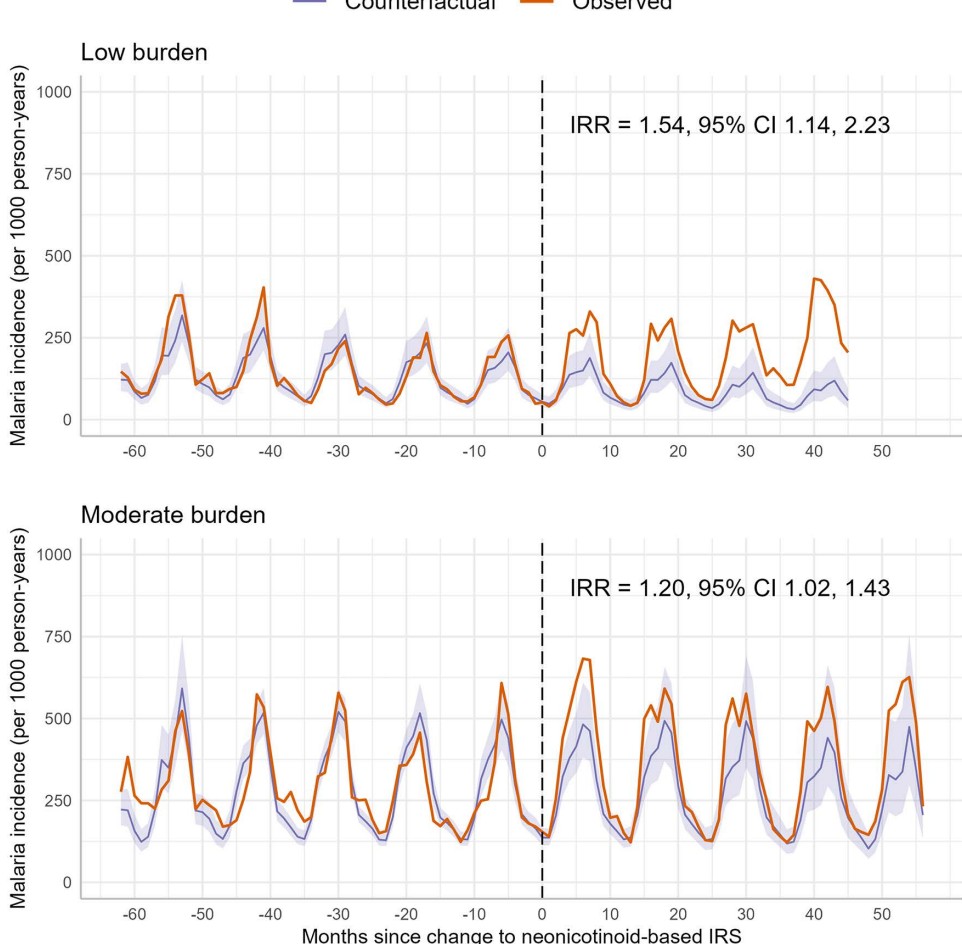

**Fig 4. Observed and modelled counterfactual monthly malaria incidence before and after the change to neonicotinoid-based IRS, pooled and stratified by baseline malaria incidence.** This figure presents monthly malaria incidence per 1,000 person-years before and after the switch to neonicotinoid IRS, stratified by baseline incidence (low, medium, high). Orange lines show observed DHIMS2 data; purple lines show counterfactual estimates with 95% CIs. Dashed line (Month 0) marks the switch to neonicotinoid IRS. IRRs quantify post-switch changes relative to the counterfactual after adjusting for rainfall, temperature, and NDVI.

exceeded counterfactual expectations after the intervention, with differences across models driven primarily by statistical uncertainty rather than changes in effect direction.

## Discussion

The interrupted time series revealed that replacing organophosphate insecticides with neonicotinoids in IRS campaigns across northern Ghana was associated with a 26% increase in malaria incidence, compared with a scenario in which organophosphates were maintained. This was further confirmed with the reintroduction of organophosphate IRS in the Mamprugu-Moagduri district, resulting in a decline in cases. We found that the resurgence in malaria incidence was most pronounced in regions with low baseline malaria transmission, suggesting that neonicotinoid-based IRS might have been less effective in these settings compared to organophosphates. Particularly concerning was the Upper East region, where malaria incidence increased by 4.3 times the expected level in the three IRS-targeted districts following the transition

to neonicotinoids. In contrast, areas with moderate malaria intensity showed a comparatively smaller increase, possibly because transmission levels were already elevated, making marginal changes less detectable. Although absolute malaria incidence rose in the Upper West region, where malaria intensity was highest, our analysis indicates that this trend cannot be attributed solely to the insecticide change. These findings underscore the critical need for robust, context-specific epidemiological monitoring when introducing new vector control products.

In our study, we investigated the transition of insecticides in the same regions over time, with no indication that IRS acceptance, programmatic elements or vector-specific factors changed dramatically in this period [7,46–49]. The spray coverages for the different IRS campaigns between 2015 and 2022 have ranged from 87% to 98% (above 80% target coverage [50,51]) with high spray quality each year [19,46,52]. Vector species composition in the area has remained stable over the years with *An. gambiae* s.s., the predominant vector species, co-existing with *An. coluzzii* and *An. arabiensis* [33,53,54]. Possible disruptions from the COVID-19 pandemic may have affected the provision and uptake of malaria prevention and treatment services and might account for the increased malaria incidence in 2020 and 2021 compared to the pre-COVID-19 period, as reported in Uganda [22]. However, these disruptions are unlikely to account for the heterogeneous change in malaria incidence across the study area [39]. In fact, analyses of routine surveillance data show that treatment-seeking and health facility attendance in the Northern region declined during the COVID-19 pandemic. All outpatient department visits declined by up to 27% during the second and third quarters of 2020 compared with the 2015–2019 average, coinciding with movement restrictions and reduced access to health services [55]. Declines were also reported in suspected malaria cases (−14.1%) and malaria admissions (−21.6%) at the national level during the same period [56]. Consequently, any pandemic-related bias introduced would tend to underestimate malaria burden.

Notably, we recorded reductions in absolute malaria incidence in the Kumbungu district (Northern region) and the Wa municipality (Upper West region), despite their IRS histories being similar to those of other districts within their respective regions. Furthermore, the residual bioefficacy of both SumiShield 50WG and Fludora Fusion has been reported to last between eight and eleven months, which is similar to Actellic 300CS and exceeds the duration of the malaria transmission season in the northern Savannah zone [37,57]. The impact of neonicotinoid-based IRS on malaria incidence appears to be heterogeneous and is likely influenced by the simultaneous, gradual implementation of SMC across the region. The overlap between the successful introduction and high uptake of SMC across the different regions and the shift in IRS insecticides could indicate a larger resurgence in malaria cases if SMC had not been introduced. Previous studies demonstrated the feasibility and effectiveness of SMC in reducing malaria incidence among target populations in Northern Ghana, particularly when coverage is high [58]. SMC coverage in the study area ranged from 87% to 99% of the target populations [59–61].

It could be argued that the 2016 mass ITN distribution contributed to the much superior efficacy of OP-IRS during this period due to the additional personal protective effect of a new net [62]. However, evidence from our study area suggests that pyrethroid-only ITNs that were distributed in 2016 had limited entomological and epidemiological impact in this setting. This could have been due to high pyrethroid resistance intensity [63] and low ITN utilisation (37.8% in IRS communities vs 57.3% in non-IRS communities) [64]. Analysis of entomological data shows that parity of *An. gambiae* s.l. in non-IRS districts that received ITNs increased significantly from 67% in 2016 to about 76%, following ITN distribution (Z = −7.36, p < 0.001). In districts where IRS was withdrawn in 2015, vector parity increased sharply from 28% during IRS implementation to 66% by 2017 (p = 0.001), despite the mass ITN distribution. In contrast, in the Kumbungu district, where IRS was reintroduced, parity rates declined from 55% in 2016 to 40% in 2017 (Z = 6.77, p < 0.001) [46,47,64]. Consistent with this, an observational analysis of the DHS data during this period revealed that IRS districts recorded 26% and 58% lower incidence in 2016 and 2017, respectively, than neighbouring non-IRS districts (which received only ITNs). In the Upper East Region, where IRS was withdrawn in 2015 but ITNs were provided, malaria incidence increased by an average of 485% per district (95% CI: 330–640%) relative to 2014 [7]. These observations may be explained by findings from a modelling study suggesting that even low levels of resistance would increase the incidence of malaria due to reduced

mosquito mortality and lower overall community protection [62]. The reduced impact of pyrethroid-only ITNs on vector longevity in this setting and the low utilisation of ITNs suggest that any contribution of the 2016 mass ITN distribution on malaria case reduction between 2015 and 2018 would have been very minimal. The lower malaria incidence recorded during the OP-IRS period can therefore be attributed to the efficacy of pirimiphos methyl that was sprayed.

The heterogeneity in the impact of the IRS insecticide change is also likely influenced by the susceptibility profiles of local *Anopheles* vector populations. Although the use of neonicotinoids for vector control was novel in Ghana at the time of introduction, this insecticide had already been widely used in agriculture in various formulations. In fact, as of November 2019, the Ghana Environmental Protection Agency (EPA) had registered over 30 neonicotinoid-based products under different trade names for agricultural purposes [65]. Crops such as cotton, maize and soybean rely heavily on neonicotinoid-based pesticides to control common pests [65–67]. As these insecticides are highly water-soluble, they readily leach into the soil and aquatic systems [68]. One study in 2015 found that 79% of the water samples from cotton-growing communities in northern Ghana contained high concentrations of neonicotinoids [69]. Given the life cycle of *Anopheles* mosquitoes in aquatic habitats, there is a high likelihood of exposure to (sub-)lethal doses of neonicotinoids in areas where neonicotinoids are used in agriculture, as has been observed in Cameroon [70,71]. The limited impact of neonicotinoids on malaria incidence in parts of the northern Savannah zone coincides with areas of intensive cotton and soybean farming [72], suggesting potential cross-resistance due to agricultural use. Similar patterns are observed in IRS districts of Uganda and Malawi involved in neonicotinoid-reliant agriculture [73,74]. Further studies are needed to explore how exposure to agricultural insecticide affect vector susceptibility and malaria control outcomes, including potential links between crop types and insecticide efficacy.

Baseline clothianidin susceptibility testing conducted in 2017 indicated that the malaria vectors in the study area were largely susceptible to clothianidin [75], even in high cotton production areas, and this susceptibility appeared to persist after neonicotinoid IRS was introduced [37,49]. Yet, entomological surveillance after the introduction of neonicotinoids showed high parity rates, indicating high survivorship of vector species. In the North East IRS districts, vector parity rate rose from 38% during the organophosphate period to 46% under neonicotinoids (2020–2022), a 21% relative increase, suggesting a higher proportion of older, potentially infectious mosquitoes. Conversely, parity rates in unsprayed areas have shown a consistent decline (from 69% to 58%) over the same period [33,37,49]. These findings, alongside the increase in malaria incidence after neonicotinoid IRS, call into question the operational sensitivity of the current WHO phenotypic susceptibility test for clothianidin [76], which may only detect intense resistance. Furthermore, the use of rapeseed oil-based surfactants in bottle bioassays may overestimate clothianidin potency [77,78], possibly explaining the lack of detected resistance at baseline. Enhanced frequency and geographic targeting of resistance monitoring are needed, especially considering the overlap between agricultural pesticide use and IRS sites. This is particularly relevant as most of the newly approved IRS products, including Vectron T500 (a.i. broflanilide), Sylando 240SC (a.i. chlorfenapyr), and Sovrenta 15WP (a.i. Isocycloseram) [79], are repurposed agricultural insecticides [80–82].

While our study provides valuable insights, it is not without limitations. This study is based on passive surveillance data at the district level, which may have introduced reporting biases that could have influenced the modelled estimates. Health-seeking behaviour data were limited to a single DHS survey, limiting our ability to identify changes in behaviour and to assess whether trends observed in passive case detection accurately reflect true malaria incidence. While DHIS2 data quality has generally improved since 2014, there remains the possibility that issues- such as incomplete or missing data, over-reporting, variable testing rates, and false positives could still bias the estimates. As data are aggregated to the district-level, important geographical variations might have been missed. The analysis itself used general factors to model malaria incidence. More studies on factors contributing to malaria incidence in the area are necessary to fully understand the relationship between variation in IRS activities and malaria incidence, especially in the Upper West region. Additionally, qualitative studies exploring community coverages, perceptions, and adherence to IRS programs could provide a more comprehensive understanding of the factors influencing the effectiveness of malaria control interventions.

## Conclusion

This study highlights the importance of understanding the impact of switching to new insecticides in IRS programs on malaria incidence to inform malaria programming decisions. Due to increasing insecticide resistance, many malaria programs have added neonicotinoid-based insecticides to their IRS programs for effective resistance management. However, recent reports of a resurgence in malaria incidence in Uganda and now Ghana, following the transition of IRS insecticides to neonicotinoids, raise concerns about the long-term sustainability of neonicotinoids for IRS in areas where they are used extensively for agriculture. Our findings highlight the need for more studies on the reasons why neonicotinoids are underperforming compared to organophosphates in certain environments and underscore the importance of tailoring malaria control strategies to specific regional contexts using detailed resistance profiles and a deeper understanding of the impact of agricultural pesticides. Malaria control programs will need to consider a more tailored strategy, integrating multiple control methods and varying insecticides based on subnational vector resistance profiles, ecological conditions and epidemiological outcomes.

## Supporting information

**S1 Fig. District-level percentage change in malaria incidence following neonicotinoid IRS in northern Ghana.**
(TIFF)

**S2 Fig. Results from interrupted time series analysis when leaving out one district (Mamprugu-Moagduri) that reintroduced organophosphates in 2020.**
(TIF)

**S3 Fig. Results from sensitivity analysis when leaving out Gushiegu and Karaga not sprayed in 2015 and 2016.**
(JPG)

**S4 Fig. Results from sensitivity analysis when ITS is restricted to the period from 2017 to 2022 when all study districts received IRS.**
(JPG)

**S5 Fig. Results from sensitivity analysis when the primary outcome is malaria cases for the ITS.**
(JPG)

**S1 Table. Summary of type of insecticides used in our study districts from 2008-2022.**
(DOCX)

**S2 Table. Timeline for the introduction of different malaria control interventions during the study period.**
(DOCX)

**S3 Table. Pre-and Post-Neonicotinoid IRS Malaria Incidence grouped by districts and region.**
(DOCX)

**S4 Table. Incidence rate ratios comparing observed malaria incidence to counterfactual malaria incidence modelled using interrupted time series methods.**
(DOCX)

**S5 Table. Associations between covariates and malaria incidence in the final model pooled across regions.**
(DOCX)

# Acknowledgments

We are grateful to the National Malaria Elimination Programme of Ghana, the regional and district staff of the Ghana Health Services for their support of this work. The critical feedback and inputs from the PMI Evolve Ghana team and AGAMal Ltd. is also greatly appreciated.

# Author contributions

**Conceptualization:** Sylvester Coleman, Christian Atta-Obeng, Otubea Owusu Akrofi, Julie-Anne A. Tangena, Keziah Malm.

**Data curation:** Sylvester Coleman, Christian Atta-Obeng, Abdul Gafaru Mohammed, Ihsan Isaka, Wahjib Mohammed, Nana Yaw Peprah, Keziah Malm.

**Formal analysis:** Sylvester Coleman, Clinton Nkolokosa, Abdul Gafaru Mohammed, Julie-Anne A. Tangena, Adrienne Epstein.

**Methodology:** Dominic B. Dery, Adrienne Epstein.

**Supervision:** Samuel Asiedu, Samuel K. Dadzie, Keziah Malm.

**Validation:** Abdul Gafaru Mohammed, Ihsan Isaka, Wahjib Mohammed, Nana Yaw Peprah, Samuel Asiedu.

**Visualization:** Clinton Nkolokosa, Adrienne Epstein.

**Writing – original draft:** Sylvester Coleman, Julie-Anne A. Tangena.

**Writing – review & editing:** Sylvester Coleman, Christian Atta-Obeng, Clinton Nkolokosa, Abdul Gafaru Mohammed, Otubea Owusu Akrofi, Ihsan Isaka, Wahjib Mohammed, Nana Yaw Peprah, Samuel Asiedu, Samuel K. Dadzie, Dominic B. Dery, Julie-Anne A. Tangena, Adrienne Epstein, Keziah Malm.

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
