## [Decision Letter · Decision Letter 0]

4 Dec 2025

PGPH-D-25-02746

Malaria Incidence Rose Following the introduction of Neonicotinoid-Based IRS in selected districts in northern Ghana: An Observational Analysis

Dear Dr. Coleman,

Thank you for submitting your manuscript to PLOS Global Public Health. After careful consideration, we feel that it has merit but does not fully meet PLOS Global Public Health’s publication criteria as it currently stands. Therefore, we invite you to submit a revised version of the manuscript that addresses the points raised during the review process.

I would like to sincerely apologise for the delay you have incurred with your submission. It has been exceptionally difficult to secure reviewers to evaluate your study. We have now received two completed reviews; the comments are available below. The reviewers have raised significant scientific concerns about the study that need to be addressed in a revision.

Please revise the manuscript to address all the reviewer's comments in a point-by-point response in order to ensure it is meeting the journal's publication criteria. Please note that the revised manuscript will need to undergo further review, we thus cannot at this point anticipate the outcome of the evaluation process.

We look forward to receiving your revised manuscript.

Kind regards,

Miquel Vall-llosera Camps

Staff Editor

Journal Requirements:

1.    Some material included in your submission may be copyrighted. According to PLOS’s copyright policy, authors who use figures or other material (e.g., graphics, clipart, maps) from another author or copyright holder must demonstrate or obtain permission to publish this material under the Creative Commons Attribution 4.0 International (CC BY 4.0) License used by PLOS journals. Please closely review the details of PLOS’s copyright requirements here: PLOS Licenses and Copyright. If you need to request permissions from a copyright holder, you may use PLOS's Copyright Content Permission form.

Potential Copyright Issues:

Figure 1: please (a) provide a direct link to the base layer of the map (i.e., the country or region border shape) and ensure this is also included in the figure legend; and (b) provide a link to the terms of use / license information for the base layer image or shapefile. We cannot publish proprietary or copyrighted maps (e.g. Google Maps, Mapquest) and the terms of use for your map base layer must be compatible with our CC-BY 4.0 license.

Reviewers' comments:

Reviewer's Responses to Questions

**Comments to the Author**

1. Does this manuscript meet PLOS Global Public Health’s publication criteria? Is the manuscript technically sound, and do the data support the conclusions? The manuscript must describe methodologically and ethically rigorous research with conclusions that are appropriately drawn based on the data presented.? Is the manuscript technically sound, and do the data support the conclusions? The manuscript must describe methodologically and ethically rigorous research with conclusions that are appropriately drawn based on the data presented.

Reviewer #1: Partly

Reviewer #2: Yes

2. Has the statistical analysis been performed appropriately and rigorously?

Reviewer #1: No

Reviewer #2: Yes

3. Have the authors made all data underlying the findings in their manuscript fully available (please refer to the Data Availability Statement at the start of the manuscript PDF file)?

The PLOS Data policy requires authors to make all data underlying the findings described in their manuscript fully available without restriction, with rare exception. The data should be provided as part of the manuscript or its supporting information, or deposited to a public repository. For example, in addition to summary statistics, the data points behind means, medians and variance measures should be available. If there are restrictions on publicly sharing data—e.g. participant privacy or use of data from a third party—those must be specified.requires authors to make all data underlying the findings described in their manuscript fully available without restriction, with rare exception. The data should be provided as part of the manuscript or its supporting information, or deposited to a public repository. For example, in addition to summary statistics, the data points behind means, medians and variance measures should be available. If there are restrictions on publicly sharing data—e.g. participant privacy or use of data from a third party—those must be specified.

Reviewer #1: No

Reviewer #2: Yes

4. Is the manuscript presented in an intelligible fashion and written in standard English?

Reviewer #1: Yes

Reviewer #2: Yes

Reviewer #1: This is an interesting analysis that attempts to assess impact of IRS insecticide changes on malaria incidence as reported in passive surveillance data. I agree with the authors that ITS is a great approach to explore this question, however I have some concerns that the complex mix of interventions applied in the study area have been over-simplified and the impact of mass ITN distribution in 2015 and SMC introduction are insufficiently discussed in the paper. This potential bias needs to be unpicked further to justify the authors’ conclusions that increases in incidence are attributable to switch from organophosphate to neonicotinoids. Specifically:

1. Authors mention that SMC began in Upper West in 2015, and was then scaled to other regions between 2016 and 2019. It would be useful to include this phased-scale up in one of the figures or tables outlining the intervention mix over time (perhaps a hatching pattern in table S1?) Furthermore, while inclusion of a covariate for SMC is mentioned on line 202, the SMC coefficient value in the fitted model is not reported. I recommend adding a table reporting the coefficient values for the final model, so that the coefficient size and direction for SMC and the environmental variables can be seen by readers.

2. A mass ITN distribution in 2015 is mentioned in the footnote to table S1, but in the manuscript (line 130) you say that ITNs are only given as part of ANC or child welfare visits. This mass distribution could have suppressed transmission in the few years after 2015, potentially over-estimating the impact of organophosphate IRS on malaria incidence. There might be a few ways to try and incorporate the mass ITN distribution in your model: with a covariate that diminishes over time, informed by any local data on ITN durability (insecticide & physical). Another alternative would be to extend your baseline period to include 1-2 years prior to the mass distribution, though I understand this could be difficult as a result of lower data quality prior to 2015. At the least, I feel that this issue should be discussed as a potential limitation.

3. The description of your primary outcome, malaria incidence, requires a bit more detail in lines 179-185. Did you adjust for changes in testing by including testing rate directly in your model? You mention three DHIS2 indicators (suspected, tested suspected and confirmed cases), but it is not clear which were used or how, and I do not think it would be possible to reproduce the analysis based on information provided.

4. How did you code Xct for the districts that were not sprayed in 2015 or 2016? Were those district-months dropped from the analysis set, or coded the same as the districts which received organophosphate IRS?

Additional suggestions:

5. Did you consider extending this analysis to a controlled interrupted time series, by including some of the districts in these regions which did not receive any IRS? This could help with disentangling the effect IRS from that of SMC and mass ITN distribution.

6. I would be interested to see a sensitivity analysis that uses just confirmed malaria cases as the outcome, rather than incidence adjusted for testing rates. However if comment #3 is adequately addressed, this may be redundant.

7. Do you have any indication of if treatment seeking for fever changed over the study period, or indeed treatment seeking more broadly? In particular, this study period includes the COVID-19 pandemic, when treatment seeking could have altered from the usual levels. All-cause outpatient attendance data in DHIS2 could help to assess this.

8. In table S2, what does * indicate?

Reviewer #2: Comments to author:

1. This is a very well written manuscript.

2. The topic is very important given the long standing burden and challenge in the control and elimination of malaria in malaria endemic countries. It will be a valuable contribution to global efforts to control and eliminate malaria.

3. Please employ tools such as grammarly to correct the grammatical, spelling and punctuation errors within this manuscript.

**Do you want your identity to be public for this peer review?** For information about this choice, including consent withdrawal, please see our Privacy Policy..

Reviewer #1: No

Reviewer #2: No

---

## [Decision Letter · Decision Letter 1]

29 Jan 2026

PGPH-D-25-02746R1

Malaria Incidence Rose Following the introduction of Neonicotinoid-Based IRS in selected districts in northern Ghana: An Observational Analysis

Dear Dr. Coleman,

Thank you for submitting your manuscript to PLOS Global Public Health. After careful consideration, we feel that it has merit but does not fully meet PLOS Global Public Health’s publication criteria as it currently stands. Therefore, we invite you to submit a revised version of the manuscript that addresses the points raised during the review process.

Reviewer 1 has identified some errors in the supplementary tables which need correction.

We look forward to receiving your revised manuscript.

Kind regards,

Jen Edwards

Staff Editor

Journal Requirements:

Additional Editor Comments (if provided):

Reviewers' comments:

Reviewer's Responses to Questions

**Comments to the Author**

Reviewer #1: (No Response)

Reviewer #2: All comments have been addressed

publication criteria? Is the manuscript technically sound, and do the data support the conclusions? The manuscript must describe methodologically and ethically rigorous research with conclusions that are appropriately drawn based on the data presented.? Is the manuscript technically sound, and do the data support the conclusions? The manuscript must describe methodologically and ethically rigorous research with conclusions that are appropriately drawn based on the data presented.

Reviewer #1: Yes

Reviewer #2: Yes

3. Has the statistical analysis been performed appropriately and rigorously?

Reviewer #1: Yes

Reviewer #2: Yes

4. Have the authors made all data underlying the findings in their manuscript fully available (please refer to the Data Availability Statement at the start of the manuscript PDF file)?

The PLOS Data policy requires authors to make all data underlying the findings described in their manuscript fully available without restriction, with rare exception. The data should be provided as part of the manuscript or its supporting information, or deposited to a public repository. For example, in addition to summary statistics, the data points behind means, medians and variance measures should be available. If there are restrictions on publicly sharing data—e.g. participant privacy or use of data from a third party—those must be specified.requires authors to make all data underlying the findings described in their manuscript fully available without restriction, with rare exception. The data should be provided as part of the manuscript or its supporting information, or deposited to a public repository. For example, in addition to summary statistics, the data points behind means, medians and variance measures should be available. If there are restrictions on publicly sharing data—e.g. participant privacy or use of data from a third party—those must be specified.

Reviewer #1: No

Reviewer #2: Yes

5. Is the manuscript presented in an intelligible fashion and written in standard English?

Reviewer #1: Yes

Reviewer #2: Yes

Reviewer #1: Thank you for the clear and detailed responses to my previous comments. I’m satisfied with the changes made and wish to congratulate the authors on a strong manuscript.

In reading through the revised manuscript, I have just two minor edits to flag:

- I suggest checking content in table S2. The colour and labels appear to be in a different order for the Upper West Region than for the other regions, which I assume is a copy/paste error in the left-most column.

- Recommend adding units for precipitation (mm?) and temperature to table S5 so that the effect sizes can be easily interpreted. You may also want to consider rescaling or standardizing precipitation so that the IRR is more interpretable.

Reviewer #2: This is a very important and well written research which will be useful in informing the control and elimination of malaria. All comments have been addressed satisfactorily.

**Do you want your identity to be public for this peer review?** For information about this choice, including consent withdrawal, please see our Privacy Policy..

Reviewer #1: No

Reviewer #2: No

---

## [Editor Report · Decision Letter 2]

4 Mar 2026

Malaria Incidence Rose Following the introduction of Neonicotinoid-Based IRS in selected districts in northern Ghana: An Observational Analysis

PGPH-D-25-02746R2

Dear Dr Coleman,

We are pleased to inform you that your manuscript 'Malaria Incidence Rose Following the introduction of Neonicotinoid-Based IRS in selected districts in northern Ghana: An Observational Analysis' has been provisionally accepted for publication in PLOS Global Public Health.

Best regards,

Julia Robinson

Executive Editor